# Impact of Plastic-Related Compounds on P-Glycoprotein and Breast Cancer Resistance Protein In Vitro

**DOI:** 10.3390/molecules28062710

**Published:** 2023-03-17

**Authors:** Matteo Rosellini, Petri Turunen, Thomas Efferth

**Affiliations:** 1Department of Pharmaceutical Biology, Institute of Pharmaceutical and Biomedical Sciences, Johannes Gutenberg University, Staudinger Weg 5, 55128 Mainz, Germany; 2Microscopy Core Facility, Institute of Molecular Biology (IMB), Ackermannweg 4, 55128 Mainz, Germany

**Keywords:** plastic, microplastic, cytotoxicity, uptake, live cell imaging, microscopy

## Abstract

Plastic in oceans degrades to microplastics and nanoplastics, causing various problems for marine fauna and flora. Recently, microplastic has been detected in blood, breast milk and placenta, underlining their ability to enter the human body with still unknown effects. In addition, plastic contains other compounds such as plasticizers, antioxidants or lubricants, whose impact on human health is also elusive. On the cellular level, two transporters involved in cell protection and detoxification of xenobiotic compounds are the ABC-transporters P-glycoprotein (P-gp, MDR1, ABCB1) and breast cancer resistance protein (BCRP, ABCG2). Despite the great importance of these proteins to maintain the correct cellular balance, their interaction with plastic and related products is evasive. In this study, the possible interaction between different plastic-related compounds and these two transporters was investigated. Applying virtual compound screening and molecular docking of more than 1000 commercially available plastic compounds, we identified candidates most probably interacting with these two transporters. Cytotoxicity and uptake assays confirmed their toxic interaction on P-glycoprotein-overexpressing CEM/ADR5000 and BCRP-overexpressing MDA-MD-231-BCRP cell lines. To specifically visualize the results obtained on the P-glycoprotein inhibitor 2,2’-methylenebis(6-tert-butyl-4-methylphenol), we performed live cell time-lapse microscopy. Confocal fluorescence microscopy was used to understand the behavior of the molecule and the consequences that it has on the uptake of the well-known substrate doxorubicin and, in comparison, with the known inhibitor verapamil. Based on the results, we provide evidence that the compound in question is an inhibitor of the P-glycoprotein. Moreover, it is also possible that 2,2’-methylenebis(6-tert-butyl-4-methylphenol), together with three other compounds, may also inhibit the breast cancer resistance protein. This discovery implies that plastic-related compounds can not only harm the human body but can also inhibit detoxifying efflux pumps, which increases their toxic potential as these transporters lose their physiological functions.

## 1. Introduction

Although plastic could quite rightly be considered as one of our greatest ever achievements, unfortunately, it is also increasingly recognized as one of our most urgent challenges. Because of its scarce biodegradability, plastic pollution could have several consequences. Plastic pollution in the ocean [1,2,3], in the environment [4,5], and on marine animals health [6,7,8] have already been extensively studied. Due to ultraviolet light, seawater, and mechanical actions, plastic is degraded into microplastic (0.1–5000 µm) or nanoplastic (0.001–0.1 µm) [9].

We define plastic as a material that is formed by different polymers [10]. In addition, there are many other compounds that are useful for its production, such as plasticizers, flame retardants, antioxidants, or lubricants.

The actual impact of plastic on human health is still under discussion [11]. Microplastic and nanoplastic can be ingested by food and water [12,13], and it has been suggested that plastic particles may pass the gastro-intestinal epithelial barrier [14] and reach the systemic circulation, until they are detectable in blood and placenta [15,16]. Moreover, the compounds used as additives could be released from plastic particles and may interact with human cells [17]. These effects are mostly still unknown; we have just started investigating whether they are harmful or not [18].

A well-established strategy to investigate the effect of toxic compounds on human cells at the molecular level represents the use of the ATP-binding cassette (ABC) transporter-based detoxification system, such as P-glycoprotein (P-gp, *MDR1*, *ABCB1*) and breast cancer resistance protein (BCRP, *ABCG2*). However, the possible toxicity of plastic additives on P-gp and BCRP has not yet been investigated. P-glycoprotein is a membrane transporter that is expressed in the intestinal mucosal membrane, kidney proximal tubule epithelia, liver and luminal brain barrier, where it protects cells against xenobiotic toxicants [19]. In physiological conditions, the expression of P-gp is involved in absorption, distribution, metabolization, and excretion (ADME) of pharmacological drugs in healthy human subjects [20]. P-gp has a broad variety of substrates, which include among others anticancer drugs, antibiotics, and HIV protease inhibitors [21,22]. Breast cancer resistance protein is distributed in the intestine, liver, kidney, and brain, and it contributes to the protection of cells against toxic xenobiotics exposure [19]. Similar to P-gp, BCRP has several substrates, such as antivirals, anticancer drugs, and antibiotics [23,24].

This investigation was designed to study different compounds used for plastic production since the impact of xenobiotic transporters such as P-gp and BCRP on them is largely unexplored. Through in silico screenings, we identified compounds interacting with these two ABC transporters from a large library of plastic-related chemicals. As a second step, the most promising candidates were tested with various in vitro assays to study their possible interaction with the two cellular detoxicants of interest. This allowed us to determine whether these compounds may eventually affect human health.

## 2. Results

### 2.1. PyRx Screening

Using PyRx 0.8 software, we screened more than 1000 compounds derived from PubChem associated with plastic. Although there are some criticisms regarding docking analyses [25], they remain useful tools for screening libraries of compounds [26,27,28]. In Figure 1A, the majority of compounds have a binding affinity on P-gp between −7.9 and −7.0 kcal/mol. Only 7% of the selected compounds were in a range from −9.0 and −13.0 kcal/mol, indicating a high affinity for P-gp. For BCRP (Figure 1B), we obtained a similar percentage for compounds with a high affinity (range between −10.0 and −19.0 kcal/mol), while for higher values, shares were rather comparable. Based on these parameters, we selected the best 120 compounds for both proteins for further analyses.

### 2.2. Molecular Docking

To investigate the binding site of the transporters more specifically, AutoDock 4.2 was used. From the results obtained, we selected only those compounds with the lowest binding energy (LBE < −8.0 kcal/mol) for both P-gp and BCRP and those which were commercially available. An exception was diisobutyl phthalate, which had an LBE higher than −8.0 kcal/mol because this compound is widely used in plastic production and has been studied in the past [29,30]. These criteria narrowed the choice to seven compounds (from now on referred to as compounds **1** to **7**; see Table 1). The selected ones belong to different classes: compounds **1** and **2** are plasticizers, compounds **3** and **4** are UV stabilizers, and compounds **5**, **6** and **7** are antioxidants.

Except for compound **2**, the P-gp binding candidates showed LBE values between −8.39 and −12.06 kcal/mol and prediction inhibition constants (pKi) lower than 701.18 nM. Most of the BCRP-binding compounds have LBE values between −9.04 and −11.60 kcal/mol and pKi values below 236.89 nM.

The visualization of the molecular dockings showed all compounds in the LBE conformation. The interactions among the seven candidate compounds and verapamil (control) and P-gp are depicted in Figure 2. The overall structure of BCRP, the selected compounds and the standard inhibitor Ko143 are illustrated in Figure 3.

### 2.3. Cytotoxicity Assay

To examined the effect of the in silico identified compounds on cell viability, we investigated drug-sensitive CCRF-CEM and P-gp-overexpressing CEM/ADR5000 cells (Figure 4). Different concentrations (0.003 to 200 μM) were tested. All resulting IC_50_ values against both cell lines obtainable in this concentration range are shown in Table 1. Compounds **1** and **2** exhibited IC_50_ values that were higher in CCRF-CEM cells than in CEM/ADR5000 cells. This is a phenomenon known as collateral sensitivity (CS) [31,32]. The resistance ratio, which was obtained by dividing the IC_50_ of the resistant cells against the IC_50_ of the sensitive one, is displayed in Table 1. By contrast, CEM/ADR5000 cells were cross-resistant to compound **6** with an IC_50_ value approximately double (36.25 ± 6.79 μM) compared to CCRF-CEM cells (17.09 ± 1.77 μM). Compounds **3** and **4** displayed an IC_50_ value of 54.73 ± 1.42 and 31.86 ± 3.35 μM, respectively, on CEM/ADR5000 and CCRF-CEM cell lines. Compound **5** showed the lowest IC_50_ value against P-gp-overexpressing (14.42 ± 2.81 μM) and the drug-sensitive cells (15.34 ± 0.21 μM). Compound **7** displayed an IC_50_ value of 16.48 ± 1.74 μM in P-gp-overexpressing cells and a value of 20.43 ± 1.08 μM in drug-sensitive cells.

The cytotoxicity was also tested on BCRP-overexpressing MDA-MB-231-BCRP and sensitive MDA-MB-231-pcDNA cell lines (Figure 4). The same concentrations of the selected compounds (from 0.003 to 200 μM) were used to perform the experiments. As shown in Table 1, compounds **2** and **4** did not inhibit both cell lines up to 100 μM. Compounds **1** and **3** exhibited IC_50_ values of 80.66 ± 3.00 and 66.30 ± 1.38 μM in MDA-MB-231-BCRP cells and of 81.58 ± 7.41 and 60.33 ± 2.58 μM in MDA-MB-231-pcDNA cells. Compounds **5**, **6** and **7** showed comparable IC_50_ values in the BCRP-transfectant cells in a range from 17.12 ± 0.46 to 19.08 ± 0.24 μM. In the mock-transfected cells, the IC_50_ values were lower for compounds **5** and **6** (16.35 ± 0.47 and 16.46 ± 0.49 μM), whereas it was higher for compound **7** (20.21 ± 1.45 μM). Cross-resistance or collateral sensitivity was not observed (Table 1).

### 2.4. P-Glycoprotein Transport Assay

The uptake of doxorubicin in P-gp-expressing CEM/ADR5000 cells was measured using flow cytometry. As shown in Figure 5A, compounds **1**, **2**, **5**, and **7** increased the uptake of doxorubicin (dox) compared to cells to doxorubicin applied alone. Compounds **3**, **4** and **6** did not affect doxorubicin uptake in all concentrations tested (IC_50_, 2 × IC_50_ and 4 × IC_50_). Treatment of P-gp expressing cells with verapamil was used as positive control. The analysis on compound **5** suggested that this compound may inhibit P-gp function compared to verapamil.

### 2.5. BCRP Transport Assay

The uptake of doxorubicin in MDA-MB-231-BCRP cells was measured using flow cytometry (Figure 5B). Compounds **1**, **3**, **5**, and **7** increased doxorubicin accumulation at all concentrations tested (0.5 × IC_50_, IC_50_ and 2 × IC_50_). Ko143 was used as positive control. These analyses suggest that these four compounds may inhibit BCRP function compared to Ko143. Compound **6** displayed no effect on doxorubicin uptake.

### 2.6. Live Cell Time-Lapse Microscopy

The live cell time-lapse microscopy was performed to visualize the uptake of doxorubicin in P-gp-expressing cells. The purpose was to confirm the flow cytometric result of compound **5**. Figure 6A shows cells at the first analysis interval (0 h) and at the end of the interval (3 h later). In addition, Figure 6B illustrates the percentage of confluence in relation to time. Cells treated with doxorubicin followed a similar trend over time to those treated with a low concentration of compound **5** (IC_50_). Moreover, our compound under examination had a higher inhibitory activity (4 × IC_50_) than verapamil, thus confirming what has already been analyzed with P-gp transport assay.

### 2.7. Confocal Fluorescence Microscopy

To visualize the localization of doxorubicin accumulation inside the cells, confocal fluorescence microscopy was performed by testing the 4 × IC_50_ concentration of compound **5**, which showed high inhibitory activity. In Figure 7, the behavior of doxorubicin in the absence (Figure 7A) or presence of compound **5** (Figure 7B) and verapamil (Figure 7C) is highlighted. Images were acquired after 3 h treatment. The images show that doxorubicin is localized in the cytosol in the absence of inhibitors, whereas in the presence of compound **5** and verapamil, it is translocated both into the nucleus and on its membrane.

## 3. Discussion

Although the role of P-gp and BCRP has already been extensively discovered as xenobiotic detoxifiers [33], their interaction with plastics, microplastic, nanoplastic and especially with related plastic production compounds is still largely unknown. Furthermore, the inhibition of P-gp by environmental pollutants has already been studied [34], but not specifically with plastic-related compounds. Even the BCRP is involved in the transport of environmental contaminants, which also act as modulators of this protein [35]. With the intensification of plastic presence in our everyday life [36] and its correlation with these types of compounds [37], it is not surprising that plastic exposure will further increase in the future. In this scenario, the release of plastic-related compounds from plastic, microplastic, and nanoplastic is still under investigation. Some studies have shown their possible emission [38,39].

In the present study, we investigated the role of the ABC-transporters P-glycoprotein (P-gp, *MDR1*, *ABCB1*) and breast cancer resistance protein (BCRP, *ABCG2*) for seven different compounds that are involved in plastic production.

Combining the uptake assay and cytotoxicity results with the resistance ratio (Table 1), we can conclude that compound **6** may be a substrate of P-gp. Instead, compound **2** might be an inhibitor but not as strong as verapamil, since an increase in activity was noted at a high concentration (4 × IC_50_) and showed a resistance ratio lower than 0.5 (Table 1). In this context, it is possible for a mixture of plastic-related compounds to show synergistic toxicity if one compound is a substrate of the ABC-transporters and another one is an efflux inhibitor. This aspect has been discussed in the literature in the context of MXR (multiple xenobiotic resistance) of seafood [20]. While ABC-transporters can help detoxify plastic compounds that are harmful to our organism, if the plastic compounds are efflux inhibitors, they can inhibit the entire detoxification capacity and increase resistance, subsequently leading to cell death, carcinogenesis, and other diseases.

We placed particular emphasis on 2,2’-methylenebis(4-methyl-6-tert-butylphenol) (i.e., compound **5**). This compound showed apparent reproductive toxicity in rats [40] and developmental toxicity in zebrafish [41]. Compound **5** inhibited P-gp. The P-glycoprotein transport assay (Figure 5) and the graphics in Figure 6B show coherent results. In both experiments, 4 × IC_50_ showed greater inhibitory activity than the golden standard verapamil. Inhibition of P-gp, which is expressed in a wide variety of normal tissues with an important excretory role [42], leads to a disruption in cellular balance. Therefore, its inhibition no longer protects cells from both harmful plastic compounds and potentially toxic xenobiotics. There are numerous xenobiotics that are extruded by P-gp [42], maintaining the correct cellular balance. Therefore, plastic compounds further increase their toxic potential with this dual activity.

These unexpected results prompted us to further investigate the inhibitory activity of compound **5** on the uptake of doxorubicin, a well-known substrate for P-gp. In this context, we used doxorubicin only as a fluorescent probe and not as an anticancer drug. Past studies have shown that doxorubicin uptake occurs differently in the presence and in the absence of a P-gp inhibitors. A lower amount of doxorubicin was detected inside the cell in the absence of inhibitors, in accordance with the detoxifier activity of P-gp [43]. On the other hand, in the presence of an inhibitor, not only did the amount of doxorubicin within the cell increase, but it also relocated from the cytosol into the nucleus [43]. In our confocal microscope analysis, we can appreciate the same doxorubicin behavior not only for the well-established inhibitor verapamil (Figure 7C), but also for the selected compound **5**, as highlighted in Figure 7B.

Furthermore, we observed that several compounds act as BCRP inhibitors. First, good binding affinities of the compounds were found for the drug-binding site of the ABC-transporter by in silico molecular docking. Of the seven compounds analyzed by cytotoxicity assay, two of them (compounds **2** and **4**) did not show toxicity in either transporter-overexpressing (MDA-MB-231-BCRP) or sensitive (MDA-MB-231-pcDNA) cells at all concentrations tested (Figure 4). The remaining ones, on the other hand, showed cytotoxicity with similar results for compounds **5**, **6**, and **7**. Since our study was focused on the possible toxicity of plastic-related molecules, it was decided to perform further analyses on those compounds that had an IC_50_ less than 100 µM. Regarding the uptake assay (Figure 5B), compound **6** did not inhibit any concentrations tested, while for the other four compounds (compounds **1**, **3**, **5** and **7**), the inhibition increased as the IC_50_ increased. Compounds without augmentation of the inhibition at increasing concentrations compared to the control inhibitor Ko143 are the same that were involved in hydrogen bonds on Thr435 in the docking analyses. Similar results on this amino acid were studied for other BCRP inhibitors [44].

## 4. Materials and Methods

### 4.1. Chemicals

Compound **1**: dicyclohexyl phthalate (CAS 84-61-7, >99%), compound **2**: diisobutyl phthalate (CAS 84-69-5, >98%), compound **3**: octrizole (CAS 3147-75-9, >98%), compound **4**: 2-(3,5-di-tert-amyl-2-hydroxyphenyl)benzotriazole (CAS 25973-55-1, >98%), compound **5**: 2,2’-methylenebis(6-tert-butyl-4-methylphenol) (CAS 119-47-1, >99%), compound **7**: 2,2’-methylenebis(6-cyclohexyl-4-methylphenol) (CAS 4066-02-8, >97%). The compounds were purchased from TCI Deutschland GmbH (Eschborn, Germany). Compound **6**: 1,1-Bis(3,5-di-tert-butyl-2-hydroxyphenyl)ethane (CAS 35958-30-6, 96%) was purchased from abcr GmbH, Karlsruhe, Germany.

### 4.2. Cell Lines

Drug-sensitive CCRF-CEM and multidrug-resistant P-glycoprotein-overexpressing CEM/ADR5000 cells were obtained from Dr. Axel Sauerbrei (Department of Pediatrics, University of Jena, Jena, Germany) and grown in RPMI 1640 medium (RPMI, 27016021, Gibco™). The multidrug resistance phenotype of CEM/ADR5000 cells has been described [45,46,47]. The maintenance of the resistance phenotype was accomplished by doxorubicin once every two weeks. Breast cancer sensitive cells MDA-MB-231-pcDNA and resistance MDA-MB-231-BCRP, were cultured in DMEM medium (DMEM, 31966021, Gibco™, New York, NY, USA). The generation of the cell lines has been described [48]. G418 disulfate salt solution (Merck, Darmstadt, Germany) was continuously added to the resistant subline to ensure the expression of BCRP. Both media were supplemented with 10% fetal bovine serum and 1% penicillin/streptomycin (15140122, Gibco™). Cells were grown at 37 °C, 90% humidity, and a 5% CO_2_ atmosphere.

### 4.3. PyRx Screening

PyRx (https://pyrx.sourceforge.io (accessed 15 April 2020) was used for preliminary screening of binding affinity against P-gp and BCRP. We screened more than 1000 compounds associated with plastic production, previously skimmed from a larger library [49], whose three-dimensional ligand structures were downloaded from PubChem (NCBI, Bethesda, MD, USA) [50] as standard data files. The crystal structure of P-gp and BCRP with good resolution were downloaded from the Protein Data Bank (http://www.rcsb.org/) [51] as PDB files (PDB code: 6QEX and 6ETI, respectively) [52,53].

### 4.4. Molecular Docking

Molecular docking was performed for the compounds with the lowest PyRx binding energies. The binding affinities of the top 120 compounds as well as the respective known inhibitors verapamil and Ko143 were calculated for P-gp and BCRP using AutoDock 4.2. The grid box was positioned around the drug binding sites of P-gp with the center of the grid box at x = 172.811, y = 166.377, and z = 160.048 with the number of grid points (npts) of 100 in x, 116 in y, and 112 in z. For BCRP, the center of the grid box was set at x = 154.964, y = 162.200, and z = 160.337 with npts of 126 in x, 106 in y, and 104 in z. The molecular docking was executed with the Lamarckian Genetic algorithm, with 250 runs and 25 Mio evaluations for each protein. Discovery Studio Visualizer software was used for visualizing the protein–ligand interactions. Parts of this research were lead using the supercomputer Mogon II and advisory services offered by Johannes Gutenberg University Mainz (hpc.uni-mainz.de), which is a member of the AHRP (Alliance for High Performance Computing in Rhineland Palatinate, www.ahrp.info) and the Gauss Alliance e.V.

### 4.5. Cytotoxicity Assay

The cytotoxicity of 7 compounds selected from in silico screening was analyzed with resazurin reduction assay. In total, 10^4^ CCRF-CEM or CEM/ADR5000 cells and 10^4^ MDA-MB-231-pcDNA and MDA-MB-231-BCRP cells were seeded per well into 96-well plates. Cells were treated with different concentrations of the selected compounds with a range from 0.003 to 200 µM in total volume of 200 µL for 72 h. Then, 20 µL/well of resazurin 0.01% *w*/*v* (Sigma Aldrich, Taufkirchen, Germany) was added. The fluorescence intensity was evaluated with the Infinite M200 Pro-plate reader (Tecan, Crailsheim, Germany). Dose–response curves were engendered for three independent experiments for each compound, and 50% inhibition concentrations (IC_50_) were calculated. The analysis was represented using Prism 6 GraphPad Software (La Jolla, CA, USA).

### 4.6. P-Glycoprotein Transport Assay

Doxorubicin (University Hospital Pharmacy, Mainz, Germany) is a known substrate for P-gp. The uptake of doxorubicin with and without the selected compounds was performed using flow cytometry. CEM/ADR5000 cells (10,000 cells/well) were treated with 20 µM doxorubicin for 3 h. In parallel, doxorubicin treatment was combined with IC_50_, 2 × IC_50_, and 4 × IC_50_ for each compound. Verapamil (100 µM) was used as positive control for P-gp inhibition. As negative control, unstained cells were used. As positive control for doxorubicin uptake, non-P-gp-expressing CCRF/CEM cells, was used. An excitation wavelength of 488 nm and emission of 530 nm were selected. Measurements were performed by using a BD Accuri™ C6 (Becton-Dickinson, Heidelberg, Germany).

### 4.7. BCRP Transport Assay

MDA-MB-231-BCRP cells (5000 cells/well) were treated with 1 µM doxorubicin. In parallel, doxorubicin treatment was combined with 0.5 × IC_50_, IC_50_ and 2 × IC_50_. Doxorubicin fluorescence was measured using a BD LSRFortessa™ Cell Analyzer (Becton-Dickinson, Heidelberg, Germany) using blue laser at excitation 488 nm and emission 610/20 nm. Ko143 (50 nM) was used as positive control for BCRP inhibition. As negative control, unstained cells were used. As positive control for doxorubicin uptake, non-BCRP-expressing MDA-MB-231-pcDNA cells were used.

### 4.8. Live Cell Time-Lapse Microscopy

CEM/ADR5000 cells were treated in Ibidi 8-well chamber slides in the absence and in the presence of 20 µM doxorubicin for 3 h. In parallel, doxorubicin treatment was combined with IC_50_, 2 × IC_50_ and 4 × IC_50_ for the selected compound. Verapamil (100 µM) was used as positive control for P-gp inhibition. Live cell time-lapse microscopy was performed using a AF7000 widefield microscope (Leica Microsystems, Wetzlar, Germany) equipped with a Hamamatsu-Flash4-USB3-101292 camera, a LED lamp (Sola light engine, SE 5-LCR-VB, Lumencor (Portland, OR, USA)). Images were acquired with a HC PL FLUOTAR L 20×/0.40NA objective lens. In addition to Brightfield channel, images visualizing doxorubicin fluorescence were acquired using an N3 filter cube (excitation filter BP 546/12, dichroic mirror FT565 and emission filter BP 527/30), and DAPI was imaged using an A4 filter (BP 360/40, FT400, BP 470/40). Time-lapse imaging was set up so that images were acquired in each sample well every 5 min for 3 h in total. The microscope incubator temperature was set to 37 °C and 5% CO2.

### 4.9. Confocal Fluorescence Microscopy

CEM/ADR5000 cells were treated in Ibidi 8-well chamber slides with 20 µM doxorubicin for 3 h. In parallel, doxorubicin treatment was combined with 4 × IC_50_ for compound **5**. Verapamil (100 µM) was used as positive control for P-gp inhibition. After 3 h, cells were washed with PBS twice to remove doxorubicin remaining in culture medium. Incubator temperature was at 37 °C, 90% humidity and a 5% CO_2_ atmosphere. Confocal imaging was performed using STELLARIS 8 FALCON (Leica Microsystems, Wetzlar, Germany) confocal system equipped with White Light Laser (WLL). Images (2048 × 2048 pixel format, pixel size 68 nm) were acquired using 561 nm excitation line via 63×/1.40NA oil immersion lens, and the emission band (571–679 nm) was detected using a detector HyD S. Brightfield channel was imaged simultaneously using a photomultiplier tube for transmitted light. For each sample, a z-stack of 45 slices with step interval of 300 nm was acquired. Images were processed with LIGHTNING™ adaptive deconvolution (Leica) using default settings (embedding medium set to “water”).

### 4.10. Data Analysis

#### 4.10.1. Quantification of Cell Area (Cell Confluency)

Cell area was segmented from the brightfield images of the live cell time-lapse images using Ilastik machine learning (V 1.3.3post3) based pixel classification [54]. Training was performed for the whole image set. Ground truth labels were given for both background and cell class until the classifier performed the cell segmentation visually in satisfactory quality. Then, 50% probability from the probability map was used as segmentation threshold. Confluency was subsequently calculated as the number of cell pixels divided by the number of pixels in the image.

#### 4.10.2. Quantification of the Doxorubicin Plus Cells Area (Dox + Cells Confluency)

Fluorescence illumination inhomogeneity was corrected using the “Normalize Local Contrast” plugin in Fiji [55] that normalizes the contrast based on per-pixel mean and standard deviation (block radius 10 pixels, standard deviation 5 pixels). The normalized fluorescence images were loaded into Ilastik, and the dox + cells were segmented over the background using pixel classification (see Appendix A as example of the workflow). The confluency was subsequently calculated as the number of cell pixels divided by the number of pixels in the image.

#### 4.10.3. Quantification of Fractional Confluency of Doxorubicin Plus Cells

For each frame, the fractional confluency was calculated as “dox + confluency/all cell confluency.” However, the same cell has different areas in brightfield segmentation and in dox + segmentation. Thus, a correction is required for fractional confluency analysis. The correction factor was obtained from an area that had only dox + cells (see Appendix A for details). Briefly, we focused on the area with three dox + cells and calculated the corresponding confluence values. The ratio here corresponds with the real 100% fractional confluency of dox + cells.

## 5. Conclusions

The presence of these plastic-related compounds in nature is bound to increase, as is the exposure of humans with all the issues that may entail. To the best of our knowledge, this is the first study to investigate the interactions of different plastic compounds on P-gp and BCRP. In this work, we observed the strong inhibition activity of 2,2’-methylenebis(4-methyl-6-tert-butylphenol) on P-gp. This result was highlighted initially by uptake assay and was then confirmed using live cell time-lapse microscopy and confocal fluorescence microscopy. For the remaining compounds, putting together cytotoxicity and uptake analyses, we were able to show that compound **6** was a substrate of P-gp, compound **2** had a weak inhibitory activity on the transporter, and the other compounds had no inhibition. Regarding BCRP, preliminary investigations showed that for the five compounds analyzed by uptake assay, there were four possible inhibitors, while compound **6** was found to have no activity. There are still very few studies on the possible toxicity of plastics on humans, especially on those involving compounds used to produce it. Future in vivo research on plastics and related compounds and further investigations on other connected categories are needed to clarify and fully understand their impact on human health. In addition to scientific study, to mitigate the huge problem of plastic pollution, it would be useful to control plastic in waste and implement general policy measures (such as bans or prevention actions) and strategic plans to improve recycling. Until now, plastic pollution has been considered a remote environmental problem that has only affected oceans and marine animals; however, it is rapidly starting to become a threat to human health as well.

## Figures and Tables

**Figure 1 molecules-28-02710-f001:**
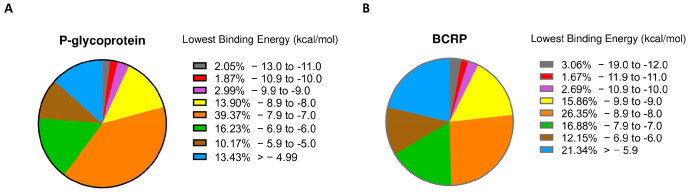
PyRx docking. Pie charts illustrating the percentage of compounds within a specific range of lowest binding of P-glycoprotein (**panel A**) and BCRP (**panel B**).

**Figure 2 molecules-28-02710-f002:**
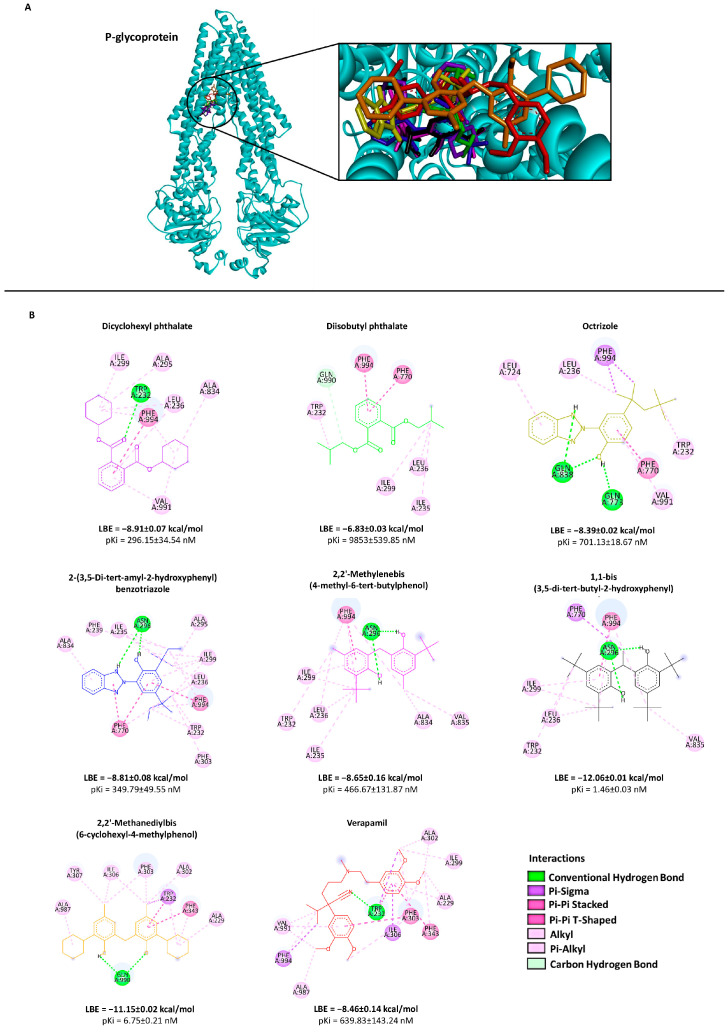
Representation of the binding mode between seven selected compounds and P-gp. (**A**) Three-dimensional structure of a P-gp model (cyan) and the lowest-energy conformation of the selected compounds and the positive control verapamil docked onto the P-gp drug-binding pocket. (**B**) Two-dimensional representation of the different types of interactions formed between the predicted interactive amino acids of P-gp and the respective selected compounds as visualized by Discovery Studio Visualizer V 21.1.0.20298 (Dassault Systemes Biovia Corp, San Diego, CA, USA). The lowest binding energies (LBE) as well as the predicted inhibition constant (pKi) values for each compound with P-gp are shown based on the molecular docking results obtained from AutoDockTools V 1.5.6 (The Scripps Research Institute, La Jolla, CA,USA). Verapamil was used as a positive control. Chemical structures are displayed according to the color code in panel (**A**).

**Figure 3 molecules-28-02710-f003:**
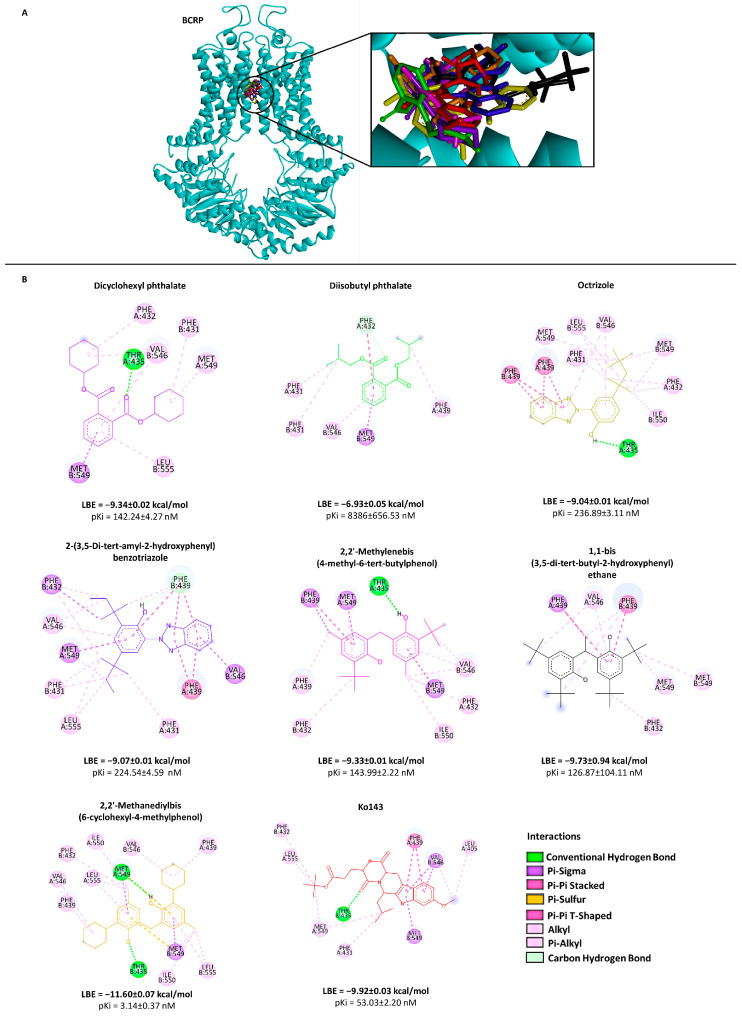
Representation of the binding mode between seven selected compounds and BCRP. (**A**) Three-dimensional structure of a BCRP model (cyan) and the lowest-energy conformation of the selected compounds and the positive control Ko143 docked onto the BCRP drug-binding pocket. (**B**) Two-dimensional representation of the different types of interactions formed between the predicted interactive amino acids of BCRP and the respective selected compounds as visualized by Discovery Studio Visualizer software. The lowest binding energies (LBE) as well as the predicted inhibition constant (pKi) values for each compound with BCRP are shown based on the molecular docking results obtained from AutoDockTools. Ko143 was used as a positive control. Chemical structures are displayed according to the color code in panel (**A**).

**Figure 4 molecules-28-02710-f004:**
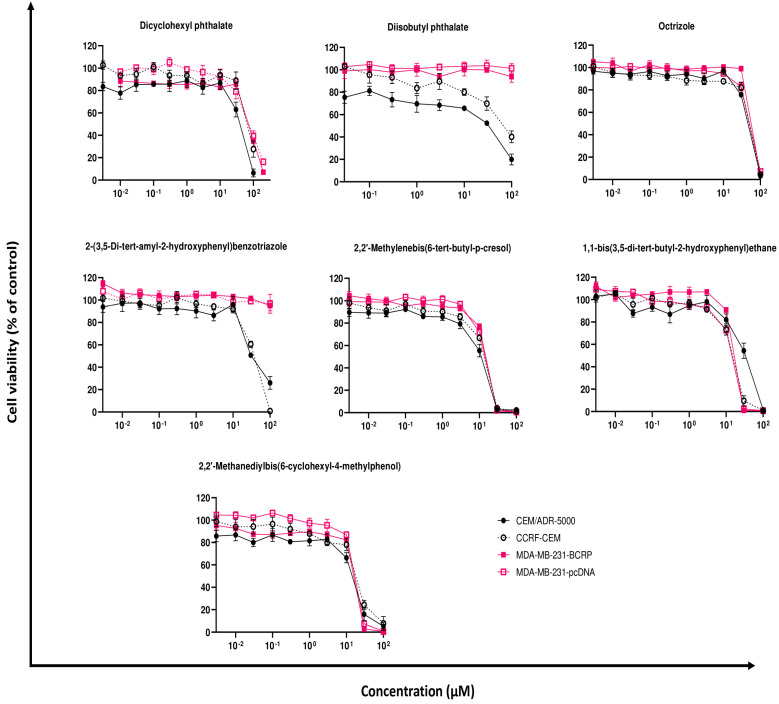
Resazurin assay of drug-sensitive CCRF-CEM and MDA-MD-231-pcDNA and multidrug-resistant CEM/ADR5000 and MDA-MD-231-BCRP treated with different concentrations of the 7 selective compounds. All experiments were performed in three replicate measurements of mean ± SD.

**Figure 5 molecules-28-02710-f005:**
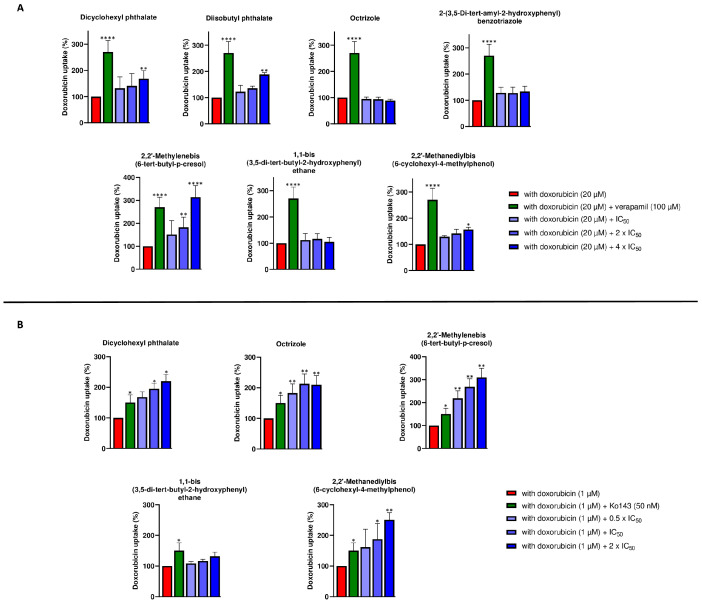
P-gp and BCRP uptake assay. (**A**) Quantification of fluorescence intensity as bar diagram on CEM/ADR5000-overexpressed P-gp cells. Experiments were performed in three replicates and are depicted as mean ± SD. (* *p* < 0.05, ** *p* < 0.01, **** *p* < 0.0001, compared to doxorubicin-treated cells). CEM/ADR5000 cells treated with doxorubicin (red) and doxorubicin in combination with three different concentrations (IC_50_, 2 × IC_50_, 4 × IC_50_) for seven selected compounds (scale of blue). Cells treated with doxorubicin in combination with verapamil, a known P-gp inhibitor, which was used as positive control (green). (**B**) Quantification of fluorescence intensity as bar diagram on MDA-MB-231-BCRP-overexpressed BCRP cells. Experiments were performed in three replicates and are depicted as mean ± SD. (* *p* < 0.05, ** *p* < 0.01, compared to doxorubicin-treated cells). Cells treated with doxorubicin (red) and doxorubicin in combination with three different concentrations (0.5 × IC_50_, IC_50_, 2 × IC_50_) for five selected compounds (scale of blue). Cells treated with doxorubicin in combination with Ko143, a known BCRP inhibitor, which was used as positive control (green).

**Figure 6 molecules-28-02710-f006:**
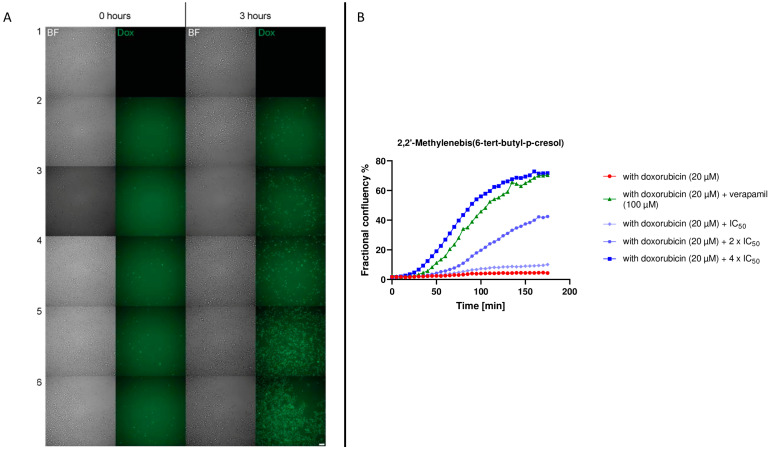
Live cell time-lapse microscopy. (**A**) Snapshots of Brightfield channel (BF) and doxorubicin fluorescence (Dox) from the beginning (0 h) and at the end (3 h) for CEM/ADR5000 cells with different conditions (samples 1–6): (1) only medium; (2) + 20 µM doxorubicin; (3) + 20 µM doxorubicin + 100 µM verapamil; (4) + 20 µM doxorubicin + IC_50_ compound 5; (5) + 20 µM doxorubicin + 2 × IC_50_ compound **5**; (6) + 20 µM doxorubicin + 4 × IC_50_ compound 5. Scalebar: 20 µm. (**B**) Fractional confluence % in relation to time of compound **5**.

**Figure 7 molecules-28-02710-f007:**
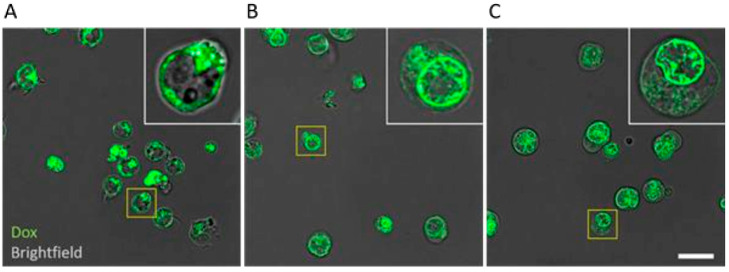
Confocal fluorescence microscopy of the middle section of the cells (overlay of the dox channel in green and Brightfield in gray) of CEM/ADR5000 cells treated with (**A**) + 20 μM doxo, (**B**) + 20 μM dox + 4 × IC_50_ compound 5 and (**C**) + 20 μM doxo + 100 μM verapamil. Scalebar is set to 20 µm.

**Table 1 molecules-28-02710-t001:** IC_50_ values of seven selected compounds on P-gp-overexpressing CEM/ADR5000 cells, sensitive CCRF-CEM cells, BCRP-overexpressing MDA-MB-231-BCRP cells and sensitive MDA-MB-231-pcDNA cells and resistance ratio of P-glycoprotein- and BCRP-overexpressing cell lines. The data are shown as mean values ± SD of three independent experiments. The resistance ratio was calculated by dividing the IC_50_ value of the resistant cell line with the IC_50_ value of the corresponding sensitive cell line.

Compound	Structure	CEM/ADR5000 IC_50_ (µM)	CCRF-CEM IC_50_ (µM)	Resistance Ratio P-gp	MDA-MB-231-BCRP IC_50_ (µM)	MDA-MB_231-pcDNA IC_50_ (µM)	Resistance Ratio BCRP
**(1) Dicyclohexyl phthalate**	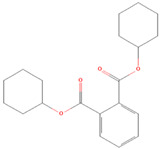	45.76 ± 7.38	73.03 ± 10.10	0.63	80.66 ± 3.00	81.58 ± 7.41	0.99
**(2) Diisobutyl phthalate**	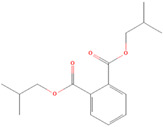	35.13 ± 4.77	75.15 ± 8.57	0.47	>100	>100	/
**(3) Octrizole**	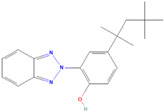	54.73 ± 1.42	58.82 ± 2.69	0.93	66.30 ± 1.38	60.33 ± 2.58	1.10
**(4) 2-(3,5-Di-tert-amyl-2-hydroxyphenyl)benzotriazole**	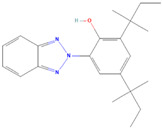	31.86 ± 3.35	40.20 ± 3.05	0.79	>100	>100	/
**(5) 2,2′-Methylenebis(4-methyl-6-tert-butylphenol)**	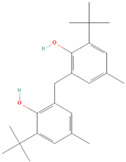	14.42 ± 2.81	15.34 ± 0.21	0.94	17.12 ± 0.46	16.35 ± 0.47	1.05
**(6) 1,1-bis(3,5-di-tert-butyl-2-hydroxyphenyl)ethane**	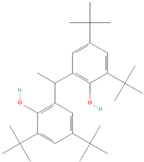	36.25 ± 6.79	17.09 ± 1.77	2.12	19.08 ± 0.24	16.46 ± 0.49	1.16
**(7) 2,2′-Methanediylbis(6-cyclohexyl-4-methylphenol)**	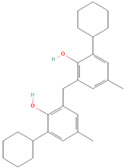	16.48 ± 1.74	20.43 ± 1.08	0.81	18.12 ± 0.21	20.21 ± 1.45	0.90

## Data Availability

Data are available upon reasonable request.

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
