# Peer review of "Impact of Plastic-Related Compounds on P-Glycoprotein and Breast Cancer Resistance Protein In Vitro"

_molecules, 2023, doi:10.3390/molecules28062710_

Round 1
Reviewer 1 Report
This is an outstanding paper on an important topic. It should attract significant media attention. I hope that the authors' institutions Public Relations (PR) departments are up to that task.
The paper has one problem that affects readability. It uses far too many abbreviations that may not be familiar to the reader. Yes, many of the abbreviations are clearly identified in the Introduction section. But it is still a problem, especially if it is read by non-specialists.
For instance, the abbreviation P-gp appears on line 177, but then the word P-glycoprotein appears on the same page, on line 203. Are they not the same thing? One way to fix this is to provide a simple list of abbreviations.
Author Response
This is an outstanding paper on an important topic. It should attract significant media attention. I hope that the authors' institutions Public Relations (PR) departments are up to that task.
The paper has one problem that affects readability. It uses far too many abbreviations that may not be familiar to the reader. Yes, many of the abbreviations are clearly identified in the Introduction section. But it is still a problem, especially if it is read by non-specialists.
For instance, the abbreviation P-gp appears on line 177, but then the word P-glycoprotein appears on the same page, on line 203. Are they not the same thing? One way to fix this is to provide a simple list of abbreviations.
RE: List of abbreviations was added to the manuscript
Reviewer 2 Report
The manuscript "Impact of plastic compounds on P-glycoprotein and breast cancer resistance protein in vitro" describes the investigation of interactions of 7 plastic-related compounds with two xenobiotic transporters, P-glycoprotein (Pgp) and breast cancer resistance protein (BCRP). While the title mentions "plastic compounds", it is probably more "plastic-related compounds", so at first I would encourage the authors to correct the title of the manuscript.
I did not notice any problems with the experimental part of the study, however the in silico investigation part has multiple issues and has to be improved before publication.
First, some general remarks about the manuscript:
* The text is a bit hard to comprehend, possibly because the Methods part is in the end, after the results description. I would suggest the authors to include some general workflow description in the beginning of the Results part, if the Methods are described in the end.
* Table 1 now seems to be out of the context, it should be near the text where the data in table are described. I would also suggest to add structural formulas of the compounds into the table, it would make the article more easy to comprehend.
* The results or BCRP are not analyzed in much detail. Probably it is because most of investigated compounds did not interact with this transporter, but the reasons could be stated more clearly.
* Clearer conclusions are necessary: after reading the manuscript, I did not understand which compunds do inhibit Pgp or BCRP and in which assays? A summarizing table would be very nice.
Now, the issues related to the computational part of the study:
* I did not understand how were the 1000 compounds from PubChem selected?
* After reading the whole manuscript and some additional googling, I understood that both PyRx and AutoDock performed docking. The authors present these two methods as very distinct, but in fact they are similar. This raises several questions, for examples, if the results were the same different from these two docking methods? Also, it would be interesting to know more details how PyRx predicts the binding affinities.
* I would not trust the predicted protein-ligand affinities, it is known since long time ago that they are not accurate (Warren et al., 2006, https://doi.org/10.1021/jm050362n).
* Fig. 2 and Fig. 3 are not clear, the resolution is not sufficient to understand them. It is important to see which residues interact with the ligands, and to what domain of the large proteins are they bound. For example, if they bind to ATP-ase domain, the inhibition is not related to transport, etc.
* I did not understand how are the values in scatter plots in Fig. 1 obtained? Are these experimentally measured or predicted inhibition constants and binding energies? The description is missing. If both are estimated computationally (predicted), then these scatter plots and high correlation are meaningless.
* Docking is not considered to be a good method to analyze ligand binding to such broad specificity transporters or enzymes (Pgp, BCRP, CYP3A4, hERG, to name a few). For Pgp substrate specificity, simple physicochemical rules have been proposed (Didziapetris et al., 2003, https://doi.org/10.1080/10611860310001648248), I do not know if something similar was proposed for BCRP. Probably using such a simple filtering would helpful during filtering as well as for interpretation of docking results.
* The articles describing the experimental structures of proteins are not cited, only the PDB IDs are given. I am also interested how these structures were selected from many available structures of the transporter proteins (Pgp and BCRP)?
To summarize: the computational part of the study has many issues and raises multiple questions. It even seems that the authors first investigated some compounds and then did an in silico study just to justify their choice. This part has either to be revised, or probably could be just removed from the manuscript, making it clearer and easier to comprehend.
Author Response
The manuscript "Impact of plastic compounds on P-glycoprotein and breast cancer resistance protein in vitro" describes the investigation of interactions of 7 plastic-related compounds with two xenobiotic transporters, P-glycoprotein (Pgp) and breast cancer resistance protein (BCRP). While the title mentions "plastic compounds", it is probably more "plastic-related compounds", so at first I would encourage the authors to correct the title of the manuscript.
RE: the title was modified
I did not notice any problems with the experimental part of the study, however the in silico investigation part has multiple issues and has to be improved before publication.
First, some general remarks about the manuscript:
- The text is a bit hard to comprehend, possibly because the Methods part is in the end, after the results description. I would suggest the authors to include some general workflow description in the beginning of the Results part, if the Methods are described in the end.
RE: the journal requires in the "Research Manuscript Sections" to first put the results and discussion and then the methods and materials. Many of the articles published in this journal are developed in this way.
- Table 1 now seems to be out of the context, it should be near the text where the data in table are described. I would also suggest to add structural formulas of the compounds into the table, it would make the article more easy to comprehend.
RE: table 1 was moved closer to the text and structures were added
- The results or BCRP are not analyzed in much detail. Probably it is because most of investigated compounds did not interact with this transporter, but the reasons could be stated more clearly.
RE: the results of the BCRP were better described in the discussion.
The analysis on the BCRP stopped at the uptake assay as the microscopic analysis is much more longer and expensive. From these preliminary analyses, potentially 4 possible inhibitors were found.
- Clearer conclusions are necessary: after reading the manuscript, I did not understand which compunds do inhibit Pgp or BCRP and in which assays? A summarizing table would be very nice.
RE:: a new conclusion was added
Now, the issues related to the computational part of the study:
- I did not understand how were the 1000 compounds from PubChem selected?
RE: The library of compounds was already present in our database as it was used by members of our laboratory for past research. This library was taken using various sites such as:
https://www.cdc.gov/biomonitoring/environmental_chemicals.html,
https://comptox.epa.gov/dashboard/chemical-lists/CPPDBLISTB
and others.
For this research, this library was taken, which contains both compounds related to plastics for its production, as well as environmental pollutants and other compounds. Only compounds related to plastic production were taken and about 1000 were found. The structures of these 1000 compounds were downloaded from the PubChem site.
- After reading the whole manuscript and some additional googling, I understood that both PyRx and AutoDock performed docking. The authors present these two methods as very distinct, but in fact they are similar. This raises several questions, for examples, if the results were the same different from these two docking methods? Also, it would be interesting to know more details how PyRx predicts the binding affinities.
RE: PyRx was used as a rapid screening program to identify possible candidates from a comprehensive chemical library. AutoDock was used as a second independent programme to verify the data obtained from PyRx and study the binding site of molecules in detail. In addition, AutoDock can visualise interactions and generate figures. This is not possible with PyRx.
- I would not trust the predicted protein-ligand affinities, it is known since long time ago that they are not accurate (Warren et al., 2006, https://doi.org/10.1021/jm050362n).
RE: Docking analysis has its disadvantages, but it remains an excellent method for skimming very large libraries of compounds. On this point, a part was added in the article.
- 2 and Fig. 3 are not clear, the resolution is not sufficient to understand them. It is important to see which residues interact with the ligands, and to what domain of the large proteins are they bound. For example, if they bind to ATP-ase domain, the inhibition is not related to transport, etc.
RE: the resolution of the figures was improved. Unfortunately, the figures are inserted directly into the text, so the quality slightly drops when this happens. However, a folder with the figures and tables in their original version has also been sent.
- I did not understand how are the values in scatter plots in Fig. 1 obtained? Are these experimentally measured or predicted inhibition constants and binding energies? The description is missing. If both are estimated computationally (predicted), then these scatter plots and high correlation are meaningless.
RE: the scatter plots was removed from figure 1
- Docking is not considered to be a good method to analyze ligand binding to such broad specificity transporters or enzymes (Pgp, BCRP, CYP3A4, hERG, to name a few). For Pgp substrate specificity, simple physicochemical rules have been proposed (Didziapetris et al., 2003, https://doi.org/10.1080/10611860310001648248), I do not know if something similar was proposed for BCRP. Probably using such a simple filtering would helpful during filtering as well as for interpretation of docking results.
RE: As already mentioned, docking analysis has its disadvantages, but it remains an excellent method for skimming very large libraries of compounds. Moreover, as docking is only a preliminary analysis, we did not rely solely and exclusively on its results to come to conclusions, but performed several in vitro tests to verify them.
- The articles describing the experimental structures of proteins are not cited, only the PDB IDs are given. I am also interested how these structures were selected from many available structures of the transporter proteins (Pgp and BCRP)?
RE: Articles on transporters were added to the specific section. The transporters were selected because they offer good resolution and have already been used in our laboratory for other past research.
To summarize: the computational part of the study has many issues and raises multiple questions. It even seems that the authors first investigated some compounds and then did an in silico study just to justify their choice. This part has either to be revised, or probably could be just removed from the manuscript, making it clearer and easier to comprehend
RE: We agree with the reviewer that computational analyses are often problematic. In our case, however, the in silico study is an integral and important part that cannot be deleted without destroying the entire design of the manuscript. We used virtual drug screening (using PyRx) as first step to identify possible chemical candidates, then we verified these candidates with a second bioinformatical method, i.e., molecular docking (using AutoDock 4.2.6). To validate these two in silico techniques we then used a in vitro assay, i.e. uptake assay. This means that our in silico-in vitro dual approach is not error-prone because it was sufficiently validated. This approach provided us reliable information which was the basis for the entire subsequent experiments. Therefore, we not like to delete the bioinformatical part and hope that the reviewer can understand our reasons.
Round 2
Reviewer 2 Report
I would like to thank the authors for the improvements of the manuscript. As a computational researcher, I really liked to improvements to the in silico part. However, I still see some minor inconsistencies. Editing the description of the in silico part would make the manuscript more easy to comprehend and thus more interesting for the readers:
* The selection procedure for the 1000 initial compounds is still not described. If the same dataset was used in some prior research that is already published, why not cite the articles where the selection procedure is probably described? If the other mentioned past studies are not yet published, then maybe writing a few sentences about the selection of the initial compounds from PubChem would be nice. It would also make the research more reproducible.
* There is a sudden jump from 120 compounds selected after PyRx analysis to cytotoxicity studies in section 2.2. Maybe docking was meant here, not cytotoxicity? This makes the beginning of section 2.2 and also the whole workflow description not very clear and could be easily improved.
* Authors mention in the reply to reviewers that the PDB structures of transporters were selected according to their good resolution. This note could be also added to the Methods part (section 4.3).
As a side note, it would be interesting to see how the compounds selected for the experimental investigation fit into the simple physicochemical model of Didziapetris et al. (https://doi.org/10.1080/10611860310001648248). I think that PubChem provides the necessary properties to do this analysis, but probably this is also outside the scope of the current manuscript.
Author Response
I would like to thank the authors for the improvements of the manuscript. As a computational researcher, I really liked to improvements to the in silico part. However, I still see some minor inconsistencies. Editing the description of the in silico part would make the manuscript more easy to comprehend and thus more interesting for the readers:
- The selection procedure for the 1000 initial compounds is still not described. If the same dataset was used in some prior research that is already published, why not cite the articles where the selection procedure is probably described? If the other mentioned past studies are not yet published, then maybe writing a few sentences about the selection of the initial compounds from PubChem would be nice. It would also make the research more reproducible.
RE: an article was added.
- There is a sudden jump from 120 compounds selected after PyRx analysis to cytotoxicity studies in section 2.2. Maybe docking was meant here, not cytotoxicity? This makes the beginning of section 2.2 and also the whole workflow description not very clear and could be easily improved.
RE: section 2.2 was modified.
- Authors mention in the reply to reviewers that the PDB structures of transporters were selected according to their good resolution. This note could be also added to the Methods part (section 4.3).
RE: section 4.3 was modified.
As a side note, it would be interesting to see how the compounds selected for the experimental investigation fit into the simple physicochemical model of Didziapetris et al. (https://doi.org/10.1080/10611860310001648248). I think that PubChem provides the necessary properties to do this analysis, but probably this is also outside the scope of the current manuscript.
RE: It would be interesting to see if the compounds fit, but as you also pointed out, this is beyond the scope of this manuscript.